# Types of Exposure Pesticide Poisoning in Jiangsu Province, China; The Epidemiologic Trend between 2006 and 2018

**DOI:** 10.3390/ijerph16142586

**Published:** 2019-07-19

**Authors:** Ning Wang, Boshen Wang, Jinbo Wen, Xiuting Li, Liping Pan, Juan Zhang, Baoli Zhu

**Affiliations:** 1School of Public Health, Southeast University, Nanjing 210009, China; 2Jiangsu Provincial Center for Disease Control and Prevention, Nanjing 210009, China; 3Department of Chronic Disease Prevention and Control, Huai’an City Center for Disease Control and Prevention, Huai’an 223001, China; 4Nanjing Prevention and Treatment Center for Occupational Diseases, Nanjing 210042, China

**Keywords:** Pesticide poisoning, Epidemiologic trend, Policy development

## Abstract

Background: Pesticide poisoning is an important issue in rural China, and is also a major public health problem that affects the health of farmers. The purpose of this paper is to explore the epidemiology of pesticide poisoning in Jiangsu Province, and to put forward the relevant suggestions on the logical and discerning utilization of the pesticides. Methods: According to the data of the pesticide poisoning report card established by the health hazard detection information system, the cases of pesticide poisoning in Jiangsu Province from 2006 to 2018 were exported to Excel, and the database of pesticide poisoning was established. Furthermore, the imported data was screened and collected. A descriptive statistical analysis had been employed on this data utilizing SPSS 20.0. Results: Between the years 2006 and 2018, 38,513 pesticide poisoning cases were registered in Jiangsu Province, with a downward trend. Meanwhile, 77.83% of poisoning cases involved insecticide poisoning, followed by herbicide and rodenticide. The greater part of the diverse sorts of studied insecticide poisoning cases involved people aged between 30 and 59 years (57.51%). Poisoning cases caused by rodenticide accounted for a large proportion of people aged between 0 and 14 years (23.72%) in non-occupational pesticide poisoning. Regarding seasons, it was distinguished that more insecticide cases were reported in autumn (46.95% of the total number of cases). Pesticide poisoning was reported in 13 cities of Jiangsu Province, and among these, insecticide poisoning was reported mainly from the northern Jiangsu area, which was the same as rodenticide and herbicide. Conclusions: Although there is a decreased risk for pesticide poisoning among farmworkers in Jiangsu Province, the number of farmworkers with pesticide poisoning is still high. Further management of the pesticide utilization is necessary, especially insecticide. More attention ought to be paid to the protection of vulnerable groups, including children and the elderly.

## 1. Introduction

China has a vast territory with different uses of pesticides in various regions [1]. As a sort of contribution in agricultural production, pesticide plays a significant role in the elimination of pests [2]. Furthermore, it is a precondition for crop protection. Pesticides such as insecticides, herbicides, fungicides, and plant growth regulating agents are considered to be cost-effective, labor-saving, as well as effective means that are necessary for the management of pests and have been used in several sectors for agricultural production [3].

Presently, there are over 2000 varieties of pesticides used across the globe. More than 600 types of pesticides are available in the markets, whereas 64 types of pesticides enjoy sales of hundreds of millions of dollars [4]. However, in the past few years, the development of the insecticide market has slowed down due to large areas of insect-resistant cotton and insect-resistant corn, leading to a direct result of the decline in the utilization of pesticides [5,6]. At the same time, herbicide-resistant crop cultivation has squeezed the selective herbicide market, thereby further reducing the development of the herbicide market [6]. Currently, chemical pesticides, particularly insecticides and herbicides, are the best for ensuring the grain and crop yield and a stable production.

The application of pesticides has significantly increased in intensity in China [7,8]. In the past few years, attention has shifted to the safety of agricultural products at home and overseas. The international and domestic institutions have taken some serious control measures to accelerate the restrictions and prohibit the production of highly toxic pesticides. Genuine concerns have been raised concerning the safety risks resulting from occupational exposure and intentional, suicidal poisoning [9], despite the fact that pesticides are produced through extremely strict control procedures in order to function with a sensible application as well as negligible effects on human wellbeing and nature. The improper and illogical utilization of pesticides also causes negative externalities. Farmers need safety precautions when using or spraying the pesticides, since the products of the soil and pesticide deposits in nature can be dangerous and might lead to a genuine danger to people’s health [10,11,12].

The occupational exposure to pesticides regularly takes place in farms [10,11,12]. It has been reported that the incidence rate of pesticide poisoning in Jiangsu Province is among the top five in China due to the high utilization of pesticides in the region, mainly for the mass agricultural production of vegetables, wheat, an rice, among others [13]. The study of pesticide poisoning in China is primarily undertaken by investigating the incidence data obtained from hospital reports. Due to this, the genuine cases of pesticide poisoning might not be acquired accurately [14]. This paper demonstrates the present cases of pesticide utilization in the Jiangsu Province, China and the potential dangers of pesticide use from the perspectives of historical trends, regional characteristics as well as the structure of pesticide utilization. The aim is to investigate the current issues and lay a theoretical foundation for the prevention of pesticide poisoning. 

## 2. Methods and Materials

### 2.1. Data Source

A report published by the “Jiangsu Provincial Center for Disease Control and Prevention” indicated Jiangsu Province’s pesticide poisoning report between 2006 and 2018. The respondents are the pesticide poisoning cases that utilized the pesticides in agricultural production and in forestry products or in the self-use of different pesticides. 38,513 cases were confirmed in this study.

### 2.2. Case Definition

From the pesticide poisoning report card, the basic characteristics of the collected pesticide poisoning patients were recorded, including information on the gender, age, region, season, type of poisoning and cause of poisoning. Pesticide poisoning mostly incorporates two noteworthy classifications: occupational as well as non-occupational. Occupational pesticide poisoning was divided into private use and employment. Private use refers to the use of pesticides by farmers for personal purposes. Employment denotes the hired farmers’ use of the pesticides, leading to poisoning. Non-occupational pesticide poisoning was divided into accidental and suicidal. Pesticides mainly included insecticides, rodenticide, herbicides and others (including bactericide, mixed formulations, and biochemical pesticide). 

### 2.3. Data Analysis

The data on pesticide poisoning in Jiangsu Province from 2006 to 2018 were collected and evaluated. An epidemiological analysis was conducted using Excel. The pesticide poisoning rate was calculated according to the patient’s age, area, type of pesticide exposure and season. Meanwhile a descriptive statistical analysis had been employed on this data, utilizing SPSS 20.0 (IBM, Armonk, NY, USA). Chi-square analyses was used to compare the composition of the pesticide poisoning population with two-tailed tests. *p*-values < 0.05 signified the statistical significance.

### 2.4. Ethics

This investigation took into consideration the use of a secondary data analysis. Patients’ personal information regarding pesticides had been encoded at the Jiangsu Provincial Center for Disease Control and Prevention (CDC). As required by ethical standards, this investigation adjusted to the Declaration of Helsinki and had also been suggested for exemption from the review of institutional ethics by the Research Ethics Board of Jiangsu Provincial CDC.

## 3. Results

An aggregate of 38,513 cases involving pesticide poisoning had been distinguished within Jiangsu Province between 2006 and 2018. The pesticides responsible for poisoning were mainly the insecticides and the herbicides. Poisoning caused by insecticides constituted 29,971 cases (77.82% of all cases), while cases involving herbicide poisoning were 3861 (10.03%). Furthermore, cases of poisoning caused by rodenticide numbered 998 (2.59%). 

### 3.1. Distribution of Pesticides Related to Pesticide Poisoning

Occupational pesticide poisoning constituted 468 cases (5.93%) in employment relationships; furthermore, there were 7423 cases of poisoning caused by private use (94.07%). In this case, pesticide poisoning was mostly caused by insecticides, which accounted for 6673 cases, or 84.56% of all the cases, followed by the herbicides and the rodenticides. In the non-occupational pesticide poisoning group, suicide caused by insecticide poisoning comprised 16,569 cases, accounting for 71.69%. Meanwhile, accidental poisoning constituted 3463 cases (14.98%). Among all of the poisoning cases, pesticide poisoning caused by insecticides comprised 29,971 cases (77.82%), comprising the vast majority. A relevant number of non-occupational poisoning cases were caused by insecticides (suicide), as illustrated in Figure 1.

### 3.2. Age Groups with Pesticide Poisoning Cases

In the occupational poisoning group, the number of insecticide poisoning cases was increasing gradually with the increase of age before the age group of 60 years, after that, the numbers were continuously decreased. The trend of herbicide poisoning was similar to that of insecticides. Out of all the cases involving non-occupational poisoning, cases of insecticide poisoning were 6694 between the age of 30 and 44 years, occupying a larger proportion. Meanwhile, in non-occupational pesticide poisoning, poisoning cases caused by rodenticide poisoning accounted for a significant proportion between the age of 0 to 14 years old (232 cases accounted for 19.03%). There was a reported increasing trend of herbicide poisoning cases with age (after the 45 to 59 years range) the all non-occupational poisoning cases (χ^2^ = 126.63, *p* < 0.001), which is the same as insecticide. Figure 2 demonstrates the pesticide poisoning by age.

### 3.3. Trend with Pesticide Poisoning

Figure 3 displays a common pattern of a decline in the quantity of poisoning cases (χ^2^ = 827.01, *p* < 0.001). In particular, in the occupational poisoning group, the number of pesticide poisoning cases (caused by insecticides and other types of pesticides) markedly declined, similar to that for non-occupational poisoning. However, in an unanticipated manner, the decline was not obvious in the cases of rodenticide poisoning in the reported non-occupational poisonings during the entire period of study.

### 3.4. Season-Related Pesticide Poisoning

Occupational pesticide poisoning was reported to mainly take place during autumn. Meanwhile, the approximate proportion of different pesticide poisoning was estimated for the third quarter: 85.98% for the insecticide, 2.59% for the herbicide, and 0.17% for the rodenticide, respectively. Regarding non-occupational insecticides poisoning, the cases were mainly concentrated in summer and fall. Figure 4 illustrates the quarterly pesticide poisoning cases.

### 3.5. Prevalent Cases of Poisoning in Northern Jiangsu

Jiangsu Province is divided into 13 administrative regions. Northern Jiangsu Province is the main agricultural development area. Geographically, insecticide poisoning made up a significant proportion of cases in northern Jiangsu (containing Xuzhou, Lianyungang, Suqian, Huaian, and Yancheng) (χ^2^ = 111.20, *p* < 0.001). In the occupational insecticide poisoning group, there are 15 cases per 100,000 people in Yancheng City and 9 cases per 100,000 people in Xuzhou City. These were the top two cases regarding the insecticide poisoning rates. The non-occupational pesticide poisoning cases were similarly significant in the northern Jiangsu area (containing the Xuzhou, Huaian and Yancheng areas). In particular, the causative agents included insecticides as well as herbicides. Among non-occupational herbicide poisonings, the number of poisoning cases in Huaian (10 cases per 100,000 people) and Xuzhou (7 cases per 100,000 people) ranks as the top two, which was different from occupational herbicides poisoning. Figure 5 summarizes this perception.

## 4. Discussion

Pesticide poisoning has been identified as a significant public health problem [15]. The major reason for its occurrence includes cases of occupational, accidental, and intentional poisoning [16,17,18,19]. Until now, pesticides are utilized on a vast scale globally, including in China [20]. In addition, they can also be related to crop cultivation in Jiangsu Province, because of their involvement in agricultural activities; hence, the use of insecticides is comparatively higher [13]. In cases of non-occupational poisoning, suicide cases caused by the insecticides were extremely high when compared to the other groups. Through an analysis of the report on pesticide poisoning from 2006 to 2018, the number of cases of non-occupational poisoning caused by insecticides decreased significantly, especially suicide cases. This finding was consistent with the distribution of poisoning cases in other provinces in China [6].

The most augmented rate was observed during the ages of 30 and 59 (55.15%). Out of all the reported occupational poisoning cases, insecticide-induced poisoning was fundamentally concentrated in the age range of 45 to 59 years. Most pesticide poisonings were insecticide poisonings. This is attributed to the fact that insecticide is the most widely pesticide used in our country, compared with other types of pesticide. In addition, most occupational cases occurred in the middle-age group, and people in this age are the main labor force in society. Besides, they are the main force of family life and, therefore, are more engaged in pesticide exposure-related work when compared to other age groups; consequently, workers were more likely to be exposed to insecticides. In the non-occupational poisoning group, there was a higher number of rodenticide poisonings in the age range of 0 to 14, when compared to herbicides and other types of pesticides. In most cases, children in this age were unaware of the poisonous substances and tended to place any substance that they picked up in their mouths [21,22]. Another reason for the high number of poisoning cases involving rodenticides is the easy access to rodenticide for household use. Simultaneously, rodenticides should also be highly toxic to humans because rodents and humans have a similar metabolism. Consequently, children in the age range of 0 to 14 accounted for a higher amount of cases of rodenticide poisoning (23.72% of 978 cases) as compared to the other age groups. Therefore, there is a requirement to implement exhaustive intervention attempts, including a suitable storage and management of insecticides at home, as well as stringent regulations for poisonous insecticides utilized as household products. Such measures will possibly reduce the acute pesticide poisoning cases among children, especially rodenticides. Meanwhile, there was also observably higher rodenticide poisoning incident rates among people aged over 60 years when compared to the young people. This might be partially because rodenticide is generally stored at home, and people could get it easily. Nonetheless, it is also reported that some of the aged people living alone or experiencing health issues tend to use insecticide when committing suicide [14,23,24,25]. All these factors have increased the probability of their pesticide poisoning. Therefore, the problem of pesticide poisoning in the elderly ought to be given consideration.

There was a descending pattern in the number of cases due to pesticide poisoning during the study period. In particular, insecticide-induced poisoning exhibited a descending pattern (*p* < 0.05) when compared to other types of pesticides. This is because insecticides have the largest amount of use in China. Therefore, the state has taken control measures against insecticides when it comes to preventing pesticide poisoning. Regarding non-occupational poisoning, there was an ascending pattern noted in 2007, facilitated by the Health Hazards Monitoring Information System that was first established in 2006. As the system improved, the number of reported poisonings also increased gradually [13,14]. In the meantime, the relevant state agencies significantly focused on controlling the hazards caused by the insecticides and herbicides, thereby lessening the control of the utilization of the rodenticides. The government should further strengthen insecticides that cause poisoning without neglecting the negative effects of rodenticides in Jiangsu Province.

Additionally, occupational pesticide poisoning has a reasonably regular accumulation, mostly in the third quarter [26]. This is because the pests in the third quarter are so severe that large amounts of insecticides are needed. The opportunities for insecticide or herbicide poisoning are increasing because of high temperatures along with pesticide volatilization. In the process of pest control, it is important to vigorously promote green control technology, scientific management and other agricultural measures, together with a physical and biological control, as well as other ecological comprehensive prevention measures. Such measures can help to reduce the utilization of chemical pesticides.

The pesticide poisoning cases of the residents in the northern region of Jiangsu Province predominated in the reported cases, accounting for 58.23% of the overall cases. It is crucial that the government strengthens the prevention of pesticide poisoning. Nantong and Yancheng are cities that focus on agriculture and pesticide production. Because of the extensive use of insecticides, farmers are more likely to be exposed to insecticides than to other types of pesticides in these areas. In the case of non-occupational pesticide poisoning, the highest frequency of insecticide poisoning incidents occurred in Xuzhou, followed by Huaian. Both cities have relatively developed crop growing areas, so herbicides are used in large quantities, and farmers are more likely to be exposed to herbicides in these two regions. Due to the low level of education, farmers are unable to obtain proper use of pesticides, especially in remote and economically underdeveloped areas; as a result, few people know how to use herbicides rationally. The lack of knowledge related to pesticide poisoning is a major factor causing pesticide poisoning in farmers.

The most important thing is that the government should use multiple interventions to repeatedly urge rural residents to change their incorrect behavior when using pesticides in order to reduce the incidence of pesticide poisoning. It is worth noting that poisonings are underreported, and that the actual number of cases of pesticide poisoning is higher than that of reported cases. The data analyzed in this study comprise reported cases, which do not represent the actual incidence of pesticide poisoning.

## 5. Conclusions

As indicated by the outcomes of this investigation, pesticide poisoning remains a significant health concern in Jiangsu Province, particularly insecticide poisoning. Proper management should be conducted, as well as production regulation in relation to the utilization of highly toxic pesticides. Moreover, significant consideration ought to be given regarding the safeguarding of susceptible groups. Taking into account the harm of insecticides to people aged 30–59 years, the packaging and toxicity of insecticides should be replaced and improved.

## Figures and Tables

**Figure 1 ijerph-16-02586-f001:**
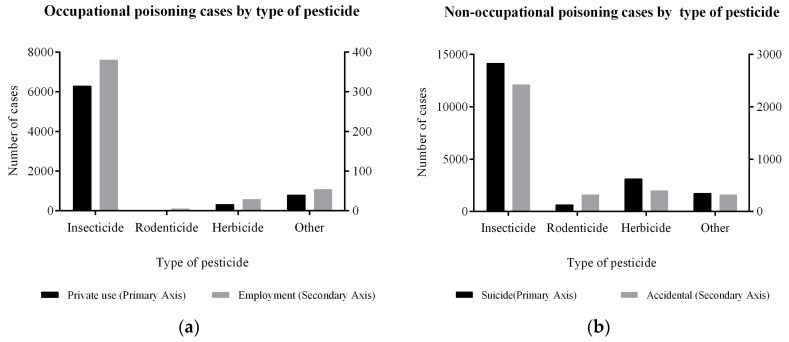
Distribution of cases due to pesticide poisoning by type of pesticide.

**Figure 2 ijerph-16-02586-f002:**
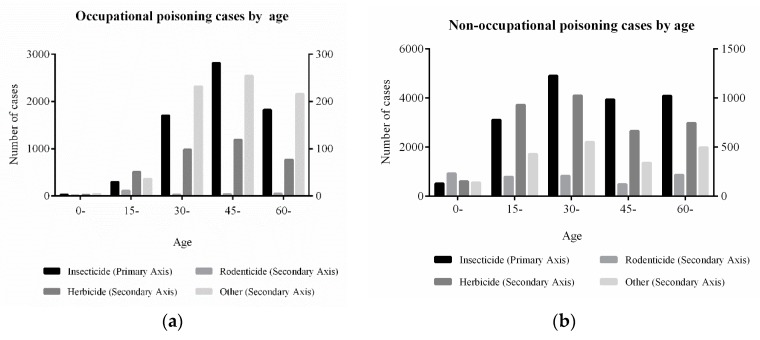
Distribution of pesticide poisoning by age.

**Figure 3 ijerph-16-02586-f003:**
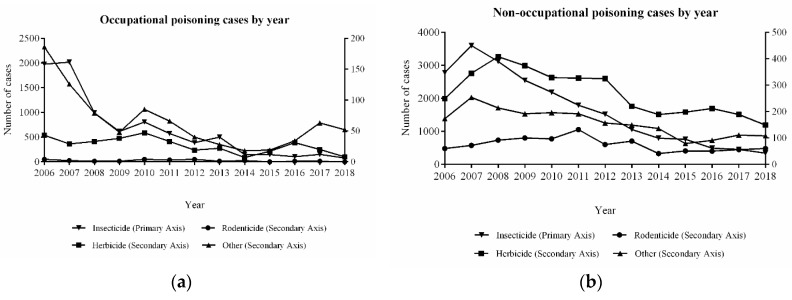
Distribution of pesticide poisoning by year.

**Figure 4 ijerph-16-02586-f004:**
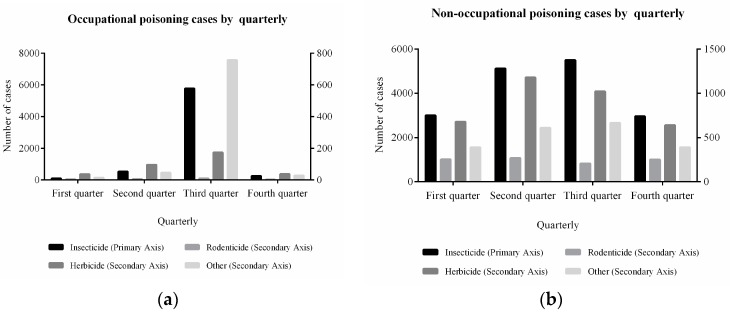
Distribution of pesticide poisoning by quarterly.

**Figure 5 ijerph-16-02586-f005:**
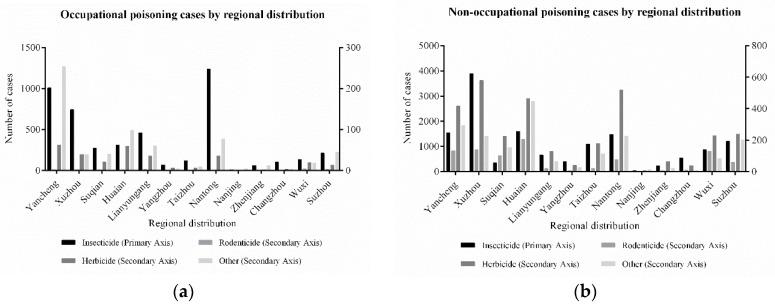
Distribution of pesticide poisoning by area.

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
