# Peer review of "Types of Exposure Pesticide Poisoning in Jiangsu Province, China; The Epidemiologic Trend between 2006 and 2018"

_ijerph, 2019, doi:10.3390/ijerph16142586_

Round 1

Reviewer 1 Report

This is an interesting study, but there are big problems with the writing of this paper.

The language is obscure, some sentences are grammatically unintelligible, and there are too many grammatical errors.

ABSTRACT-methods

The authors only describe the data entry and checking “……the case of pesticide poisoning in Jiangsu Province from 2006 to 2018 was introduced into the EXCEL, and the database of pesticide poisoning was established, the imported data was screened and collected.”but the main analyses methods should be introduced.

ABSTRACT-results

“…38513 pesticide poisoning cases were registered in Jiangsu Province, showing a downward trend.” This sentence doesn't make sense, you can either list the annual data and proportion, or change "showing" to "with"

4695% should be 46.95%, right?

Methods and Materials

2.1. Data source

"China Disease Prevention and Control Information System Occupational Disease and Occupational Health Information Monitoring System - Pesticide Poisoning Report Card". There is no need to translate the name of the report card directly, just make it clear. In addition, the translation is wrong.

2.2. Case definition

“……, the collecting pesticide poisoning patient's gender, age, region……”, the sentence is grammatically unintelligible.

2.3. Data analysis

The data analysis of this paper is too simple. It is only a descriptive analysis of the data without further data mining.

3. Results

The graphics made by EXCEL do not meet the requirements for publication, so you can use Graphpad Prism or SigmaPlot and other software to make them.

And another important question, why only 13 cities in Jiangsu province were included in the analysis?

Author Response

Response to Reviewer 1 Comments

Thanks to the editing of the manuscript and the valuable comments made by the reviewers, your suggestions will play a good role in further revising and perfecting this manuscript. We have responded to the questions and carefully revised the article as required by the reviewers. The red mark was used in the revision of the feedback article. Thank you for your opportunity to revise this manuscript.

Answers to Reviewers’ questions are as follows:

Reviewer 1:

ABSTRACT-methods

The authors only describe the data entry and checking “……the case of pesticide poisoning in Jiangsu Province from 2006 to 2018 was introduced into the EXCEL, and the database of pesticide poisoning was established, the imported data was screened and collected.”but the main analyses methods should be introduced.

Response: We supplement the main statistical methods in the manuscript abstract.

ABSTRACT-results

“…38513 pesticide poisoning cases were registered in Jiangsu Province, showing a downward trend.” This sentence doesn't make sense, you can either list the annual data and proportion, or change "showing" to "with"

Response: According to the reviewer's opinions, we made corresponding amendments in the manuscript.

4695% should be 46.95%, right?

Response: Due to our carelessness, minor errors have occurred. We have corrected the content.

Methods and Materials

2.1. Data source

"China Disease Prevention and Control Information System Occupational Disease and Occupational Health Information Monitoring System - Pesticide Poisoning Report Card". There is no need to translate the name of the report card directly, just make it clear. In addition, the translation is wrong.

Response: According to the opinions of the reviewers, we revised the content of the article accordingly.

2.2. Case definition

“……, the collecting pesticide poisoning patient's gender, age, region……”, the sentence is grammatically unintelligible.

Response: According to the purpose of the study, we re-expressed the content of the article.

2.3. Data analysis

The data analysis of this paper is too simple. It is only a descriptive analysis of the data without further data mining.

Response: Considering the opinions of the reviewers, we supplemented the statistical analysis of the constituent ratio in the article.

3. Results

The graphics made by EXCEL do not meet the requirements for publication, so you can use Graphpad Prism or SigmaPlot and other software to make them.

Response: According to the reviewer's opinion, we revised the pictures in the article.

And another important question, why only 13 cities in Jiangsu province were included in the analysis?

Response: Jiangsu Province is divided into 13 administrative regions. We analyzed the pesticide poisoning cases reported in these 13 cities.

Reviewer 2 Report

Overall, the manuscript is interesting and offer a look into pesticide-related illnesses/injuries in this region.  I have the following questions and comments:

The manuscript is not clear if the pesticide-related illnesses/injuries were associated with acute or chronic exposures.

Were there specific criteria established to determine if exposure and resultant symptoms were indeed associated? 

Though agricultural regions were mentioned, what was the proportion of agricultural use pesticides? Differentiate between pesticide handlers and bystanders.

Does lack of personal protective equipment play a role in the individual's exposure (occupational)?

What are the current regulations for agricultural use and occupational setting, and were there non-compliance that contributed to the exposure?

Has there been any outreach programs to target the small farmers on safe use?

Were there differences in the mode of exposure between occupational and non-occupational?

Note limitations of the data.

Author Response

Response to Reviewer 2 Comments

Thanks to the editing of the manuscript and the valuable comments made by the reviewers, your suggestions will play a good role in further revising and perfecting this manuscript. We have responded to the questions and carefully revised the article as required by the reviewers. The red mark was used in the revision of the feedback article. Thank you for your opportunity to revise this manuscript.

Answers to Reviewers’ questions are as follows:

Reviewer 2:

Overall, the manuscript is interesting and offer a look into pesticide-related illnesses/injuries in this region.  I have the following questions and comments:

The manuscript is not clear if the pesticide-related illnesses/injuries were associated with acute or chronic exposures.

Response: In our study, pesticide-related illnesses/injuries were associated with acute exposures.

Were there specific criteria established to determine if exposure and resultant symptoms were indeed associated?

Response: Qualified doctors from medical and health institutions identify patients based on the national diagnostic criteria for pesticide poisoning in China (GBZ8-2002, GBZ52-2002, GBZ43-2002, GBZ46-2002, GBZ34-2002, GBZ10-2002, GBZ20-2002) and information provided by patients.

Though agricultural regions were mentioned, what was the proportion of agricultural use pesticides? Differentiate between pesticide handlers and bystanders.

Response: We can't get the relative proportion of pesticide use in Jiangsu province, so we analyzed it according to the agricultural planting area.

Does lack of personal protective equipment play a role in the individual's exposure (occupational)?

Response: Due to the lack of self-protection awareness, farmers do not wear personal protective equipment when spraying pesticides, pesticides occasionally or often spill on the skin, or clothes are not washed properly when spraying, thus causing poisoning accidents of pesticide users.

What are the current regulations for agricultural use and occupational setting, and were there non-compliance that contributed to the exposure?

Response: In order to control the occurrence of pesticide poisoning incidents, the government has established a direct network reporting system and regulations for the production, sale and safe use of pesticides, as well as safety knowledge education system. Because these systems are not fully implemented, farmers lack health education and new pesticide technologies, which greatly increase pesticide exposure.

Has there been any outreach programs to target the small farmers on safe use?

Response: At present, effective measures to prevent pesticide poisoning in small farmers include advocating the use of low-toxic pesticides, strengthening publicity and education, and recommending the wearing of personal protective equipment.

Were there differences in the mode of exposure between occupational and non-occupational?

Response: The main causes of production pesticide poisoning are poor protection, long duration of operation or failure to implement the operating rules. Non-occupational pesticide poisoning is mainly caused by oral pesticides, including intentional and unintentional.

Note limitations of the data.

Response: We supplement the limitations of the data in the last paragraph of the article's results.

Round 2

Reviewer 1 Report

My main concerns have been generally solved, but the editing and writing of articles can be further improved.There are still some grammar and editing errors.

Author Response

My main concerns have been generally solved, but the editing and writing of articles can be further improved. There are still some grammar and editing errors.

Response: We revise the grammar and editing of the article accordingly